# Inheritance and Development of Self-Love: A Measurement Based on Chinese Adults

**DOI:** 10.3390/bs15050652

**Published:** 2025-05-12

**Authors:** Liming Xue, Xiting Huang

**Affiliations:** 1Business School, Yangzhou University, 196 Huayang West Road, Yangzhou 225127, China; xuepsy@126.com; 2Psychological Department, Southwest University, No. 2 Tiansheng Road, Beibei District, Chongqing 400715, China

**Keywords:** self-love, Chinese, scale development, reliability, validity

## Abstract

Self-love is a fundamental psychological construct cultivated throughout human history. In Confucian culture, it is considered the ultimate Ren, while in ancient Greek thought, it serves as the center from which love radiates. Previous qualitative research identified five dimensions of self-love, but these lacked quantitative validation. This study developed the Chinese Adult Self-Love Scale (SLS) based on prior qualitative findings, constructing an initial item bank of 90 statements. The first study assessed the item relevance and clarity, resulting in a preliminary 68-item scale. Exploratory factor analysis (EFA) of 456 participants refined it to a 22-item scale with five indicators. The second study, with 929 participants, examined its reliability and validity. Cronbach’s α exceeded 0.75, and the test–retest reliability after six weeks was 0.66. Confirmatory factor analysis (CFA) supported the scale’s validity (CFI = 0.91, TFI = 0.90, RMSEA = 0.048). This study highlights self-love’s inheritance and development among Chinese adults, as well as its cross-cultural commonalities. It provides a valid, reliable tool for measuring self-love and offers a theoretical foundation for future cross-cultural research. While limitations exist, the findings suggest promising directions for further exploration.

## 1. Introduction

Self-love has historically guided human behavior, with its philosophical foundations being rooted in diverse cultural traditions. [1] ([1]) linked self-love to friendship, “a person must first become their own good friend”, emphasizing self-friendship as a prerequisite for ethical relationships. Similarly, Immanuel [18] ([18]) viewed self-love as a reflection of human dignity and moral autonomy. In Confucianism, self-love serves a broader ethical framework, emphasizing the realization of benevolence on the foundation of self-love. For instance, Confucius posited, “Desire to have things done quickly should be your desire to have them done as you wish” ([9]). [24] ([24]) advocated to “Love others as you love yourself”, underlining that one must first cultivate oneself to better serve others and society.

Philosophical perspectives on self-love, while not requiring empirical validation, offer valuable insights that can inspire the development of psychological constructs. For instance, [29] ([29]), building upon Aristotle’s discourse on self-love, proposed that self-love comprises four components: happiness, emotions, responsibility, and excellence. [21] ([21]), influenced by Aristotle and Kant’s discussions on self-love, suggested that self-love encompasses moral virtues, self-esteem, character strengths, and stable personality traits. Within mainstream Confucian thought, Chinese scholars define self-love as cherishing one’s body, reputation, and future, accepting oneself, and refraining from self-deprecation. They view self-love as the foundation of a healthy personality ([15]; [23]; [39]).

Although self-love is deeply rooted in philosophical traditions, most psychological studies conceptualize it as a basic, common, and implicit motivation ([4]; [22]; [19]; [20]; [37]). Researchers such as [6] ([6]) used four self-relevant emotions (ashamed, humiliated, proud, pleased); they suggested that self-love was equally strong in East Asia as in North America and Western Europe. While [5] ([5]) employed the questions ‘Do you love yourself?’ and ‘How much do you love yourself?’ to measure self-love, they found that there was a significant difference between races (Caucasian, Non-Caucasian, and more than one race). The inconsistencies between the above two findings may reflect cultural and ethnic differences in measurement tools, and also challenge the conclusion that self-love is universal or culturally specific. Therefore, we need measurement tools that can take into account cultural contexts and be adapted to different social contexts to more accurately capture the multidimensional characteristics and cultural adaptation of self-love.

In comparison, within the Confucian-influenced Chinese cultural context, our previous content analysis of *Siku Quanshu* (The Complete Library of the Four Treasuries: This is a large-scale collection of documents compiled during the Qianlong period of the Qing Dynasty, covering important ancient Chinese classics. It is equivalent to a systematic collection of ancient Chinese knowledge and is regarded as an important heritage of Chinese culture.) identified three key indicators of self-love: self-cherishing, self-acceptance, and self-restraint ([36]). Additionally, in-depth interviews with contemporary Chinese individuals revealed a broader conceptualization of self-love, encompassing self-cherishing, self-acceptance, self-restraint, self-responsibility, and self-respect ([35]). Unlike the direct and individualized approaches commonly used in Western research, Chinese scholars have explored self-love in ways that align with Confucian traditions and collective societal expectations.

To bridge the gap between existing qualitative studies and psychological conceptualization, we aim to empirically validate a culturally grounded framework of self-love and operationalize it as a measurable psychological construct. We develop the Chinese Self-Love Scale (SLS) to measure how self-love reflects the lived experiences of individuals in China and how self-love manifests differently across cultural contexts. By providing a culturally informed and psychometrically sound measurement tool, this study contributes to both the theory and practice in understanding how self-love can promote mental health and flourish in diverse cultural settings.

## 2. Study 1: Development and Preliminary Measurement of Chinese SLS

### 2.1. Research Hypothesis

Based on the textual analysis of the self-love corpus in *Siku Quanshu* and the interviews with Chinese adults ([36], [35]), we propose five hypotheses, suggesting that the five main indicators of self-love are self-cherishing, self-acceptance, self-restraint, self-persistence, and self-responsibility (Table 1).

### 2.2. Item Generation and Evaluation

**Generation of the items:** The initial item pool was developed based on grounded theory, employing axial coding to identify the key themes of self-love and open coding to extract raw statements from qualitative interviews with Chinese adults ([35]). The wording of the items was directly grounded in participants’ original expressions. For example, under the dimension of self-cherishing, items such as “I cherish my health”, “I cherish my life”, and “I avoid doing things that harm my body” were adapted from interview content.

To ensure cultural and historical depth, the drafted items were then cross-checked and supplemented using modern vernacular translations of the *Siku Quanshu*. As a result, we established a culturally grounded item pool consisting of 90 self-love items.

**Evaluation of the items:** To ensure an optimum level of readability, we conducted two rounds of item evaluation (Table 2). The first round evaluated the relevance and language clarity ([11]).

### 2.3. Preliminary Test of the Chinese Adult SLS

**Recruitment and distribution:** The questionnaire’s QR code and link were generated using *[26]* ([26]) and distributed through WeChat’s Friend Circle, a popular Chinese social platform. A snowball sampling method was adopted: participants were encouraged to complete the questionnaire and then share it with others. Each participant received 3 RMB as a reward for completing the questionnaire. If they successfully referred another person who also completed the questionnaire seriously, they received an additional 1 RMB. The data collection period was from 19 February 2020 to 11 March 2020.

**Questionnaire platform and quality control:** The questionnaire consisted of items rated on a five-point Likert scale, assessing participants’ responses based on their current state. Questionnaire Star automatically randomized the order of the items for each participant to reduce order bias. To ensure data quality, two pairs of reverse-coded items were used as attention check questions (e.g., “In life, I never hurt myself” vs. “In life, I always hurt myself”). Responses with inconsistencies were excluded. The shortest recorded response time was 100 s, and the longest was 4046 s.

### 2.4. Participants

**Item assessors:** In the first round, we recruited ten participants with advanced expertise in psychology (master’s and Ph.D.) to evaluate the relevance of the sub-indicator and the item. Ten individuals with non-psychology majors participated in evaluating language clarity. The group consisted of five men and five women, with educational backgrounds ranging from high school to master’s degrees. Their ages ranged from 24 to 34 years (M = 29, SD = 3.85), and they were from cities such as Shenyang, Dalian, Harbin, Huai’an, and Lüliang. In the second round, 11 non-psychology participants evaluated language clarity again. The group consisted of four men and seven women, with educational backgrounds ranging from high school to a master’s degree. Their ages ranged from 22 to 58 years (M = 31.91, SD = 9.40). The participants were from cities such as Shanghai, Suzhou, Dalian, and Shenzhen.

**Questionnaire participants:** In total, 510 samples were collected. The questionnaires collected from three respondents under the age of 18 were deleted, and 41 respondents did not pass the attention check questions. Thus, 456 questionnaires were valid, and the effective rate was 90%. There were 86 men (18.86%) and 370 women (81.14%). Of the participants, 281 (61.62%), 36 (7.89%), 59 (21.94%), 44 (9.65%), and 32 (7.02%) were aged 18–25, 26–30, 31–40, 41–50, and 51–60 years, respectively; one person was above 60 years of age (0.22%). The largest proportion of full-time students was 255 (55.92%). Education levels varied, wherein 25 were in junior high school or below (5.48%), 29 were in high school (including technical secondary school and vocational high school) (6.36%), 61 were in college (13.38%), 308 were undergraduates (67.5%), and 33 had completed their master’s degree or above (7.24%). A total of 122 participants (26.75%) had an annual income of less than CNY 30,000; 164 people (35.96%) earned CNY 30,000–80,000; 116 people (25.44%) earned CNY 80,000–150,000; CNY 150,000–800,000; and 48 people (10.5%) earned more than CNY 800,000 (1.32%). The geographical distribution ranged from Heilongjiang, Sichuan, Liaoning, Inner Mongolia, and Xinjiang to other provinces and municipalities, such as Beijing and Tianjin, directly under the central government, as well as individuals living overseas.

### 2.5. Item Screening and Exploratory Factor Analysis

**Expert Evaluation of Item Relevance and Clarity:** To refine the initial 90-item pool, we conducted two rounds of expert evaluation focusing on item relevance and language clarity. Each item was rated on a 5-point Likert scale. For relevance, the scale ranged from 1 (“very irrelevant”) to 5 (“highly relevant”); for clarity, from “very fuzzy” to “very clear”. The average rating across the experts was used as the selection criterion.

**Exploratory Factor Analysis (EFA):** Exploratory factor analysis (EFA) was then conducted on the 68 items using SPSS 26.0, following the five-step procedure recommended by [12] ([12]). The extraction method used was Principal Component Analysis (PCA), and the rotation method applied was Varimax, which maximizes the variance of the squared loadings of a factor. The number of factors was fixed at five. Items were excluded based on the following criteria: (1) communalities being less than 0.4; (2) the absolute value of factor loadings being less than 0.4; and (3) the difference in the absolute values of factor loadings between multiple factors being less than 0.3. The analysis was repeated each time an item was eliminated until no more items met the exclusion criteria.

### 2.6. Results

**Evaluation of the items of Chinese Adult SLS:** At the end of the first round, we deleted the questions with a definition lower than 4.0 in the question matrix and modified the questions with a lower score. For the “cherish yourself” sub-indicator, the item “I cherish my family” scored 3.7 on the correlation scale. The final results showed that the average score matrix of the five items was above 4.38, and the scale had 68 items remaining (Table 3).

**EFA:** SPSS 26.0 was used to process the data, and the results showed that the Kaiser–Meyer–Olkin sampling appropriateness quantity was high (0.96). Finally, through deleting the items, Bartlett’s sphericity test results were χ^2^ = 0.88, *df* = 3285.52, *p* < 0.001. A scale with 22 items and 5 factors was obtained. The respective percentages of the five factors were: 27.90%, 10.20%, 8.68%, 6.09%, and 4.80%; they cumulatively accounted for 58.4% of the variance, and the scale had 68 items remaining (Table 4).

### 2.7. Discussion

Following the standard scale development procedures, we developed the Chinese Adult Self-Love Scale (SLS) based on qualitative research (*Siku Quanshu* and modern adult interviews) ([36], [35]). The final scale consisted of 5 indicators and 22 items, capturing both traditional and modern conceptualizations of self-love in the Chinese context. Regarding this, we will discuss two aspects: how the items and indicators of the scale met psychometric standards; and the differences between the indicators of SLS and the hypotheses of previous qualitative research.

**Psychometric Evaluation:** Initially, we conducted a preliminary evaluation of the relevance of items to the sub-indicators of self-love and the clarity of the language, resulting in a pool of self-love items. These items aligned with the item standards proposed by [10] ([10]) in psychometrics. Through EFA, and using a factor loading threshold of 0.6, we ultimately retained 22 items. The scale met basic psychometric standards.

**Differences Between the Hypothesis and the Scale:** The indicators of self-cherishing, self-acceptance, and self-restraint in the scale were consistent with the hypothesis based on the *Siku Quanshu* and modern adult interviews. In addition, the other two indicators which increased on the scale were based on the hypothesis from the modern adult interviews.

First, self-responsibility and self-respect were not retained as independent factors. Self-responsibility, which encompasses both duty and personal responsibility, has deeply permeated Chinese society as a form of “tacit knowledge” ([28]). Given its implicit nature, it may not have emerged as a distinct construct in the factor analysis. Similarly, self-persistence, previously understood as encompassing principles, dignity, boundaries, and assertiveness ([35]), has undergone significant conceptual changes. What was once considered a strict self-discipline lifestyle has now evolved into setting a positive example for others through one’s actions.

Second, two new indicators—self-kindness and self-discipline—emerged in the final scale. Self-kindness reflects individuals’ ability to set boundaries, avoid self-imposed pressure, refrain from excessive self-demands, and minimize self-embarrassment. This shift suggests a transition in modern Chinese society from an emphasis on self-respect to self-discipline, aligning with the rising awareness of self-development and psychological well-being ([41]). Similarly, the transformation of self-discipline reflects a societal shift: once associated with strict personal constraints, it has evolved into an emphasis on leading by example and positively influencing others.

## 3. Study 2: Reliability and Validity of Chinese Adult SLS

According to the results of Study 1, we tested the reliability and validity of the Chinese Adult SLS ([27]). The reliability of the scale was checked using the α coefficient and retest coefficient; for validity testing, we assessed the content validity, construct validity, and criterion-related validity.

### 3.1. Method

**Participants:** After conducting a calculation using Gpower 3.1, the questionnaire was distributed using the Questionnaire Star sample service, and the time and trap questions were once more used for quality control. Finally, 929 valid samples were recovered, and the distribution time was from 30 October 2020 to 9 February 2021. There were 407 men (43.8%) and 522 women (56.1%); 230 (24.8%) were 18–25; 182 (19.6%) were 26–30; 391 (42.1%) were 31–40; 89 (9.6%) were 40–50; 31 (3.3%) were 51–60; and 6% were older than 60 years of age. In terms of registered residence, 748 (80.5%) lived in urban and 181 (19.5%) in rural areas. Of the participants, 430 (46.3%) were only children, 411 (44.2%) had siblings, and 88 (9.5%) did not respond to this item; 241 (30.2%) were unmarried, 644 (69.3%) were married, 3 (0.3%) were divorced, and 1 (0.1%) was widowed; 27 people (2.9%) had an annual income of less than CNY 30,000; 112 people (12.1%) earned CNY 30,000–80,000; 284 people (30.6%) earned CNY 80,000–150,000; 453 (48.8%) earned CNY 150,000–500,000; 42 people (4.5%) earned CNY 500,000–1 million; and 11 (1.2%) earned more than CNY 1 million. The target sample size of the confirmatory factor analysis was determined according to the rule that the number of items must be more than 10. The minimum sample size required was 220 people, and 929 people’s answers were used.

**Scale Selection**: Criterion-related validity focused on different types of self-love and their functions. First, selecting the scale of self-love abroad as the criterion could reflect the relationship between self-love in Chinese culture and that in Western culture. We selected the subscale of self-love in the *Structural Analysis of Social Behavior* (SASB) manual compiled by [2] ([2]), and its reliability was 0.87; we also used the self-love items used by [5] ([5]).

Second, this study was based on a model of the Chinese perfect personality ([14]). Self-love, self-confidence, self-examination, self-support, and self-strength are all necessary positive self-concepts for the model of a perfect personality ([15]; [40]). Therefore, we selected the general self-confidence scale of [3] ([3]), with an internal consistency reliability of 0.92 and a test–retest reliability of 0.82. In addition, in the previous theoretical construction, responsibility for oneself was the basis of self-restraint ([35]). However, in the exploratory factor analysis, self-responsibility was hidden. Therefore, we selected the responsibility subscale of the Big Five personality traits as the criterion variable ([38]).

Third, concerning its function, self-love can affect individual and public health. Therefore, we selected the health-promoting life profile as the criterion of validity ([31]; [33]).

All questionnaires were administered to participants via the Questionnaires Stars online survey platform ([27]).

**Common Method Deviation:** To control for the common method deviation caused by having too many questions in the questionnaire, after the calculation by G*power 3.1 and after checking that all the sample sizes met our standard, we adopted three methods to control the common method deviation. First, we randomly matched questionnaires and distributed them through Questionnaire Star; for example, the Chinese Adult SLS was matched with the responsibility, general self-confidence, and health-promoting lifestyle questionnaires. We also used the partial correlation method to control for gender, age, marital status, and registered residence. Finally, all the scales were collected and tested using the Harman single-factor test ([30]). The variance percentage of the first common factor was 21.42%, less than 30% of the standard used in the current study; common method bias was, therefore, less likely to occur in this study.

### 3.2. Data Analysis

All data were imported into SPSS 26.0 and exported to the data format required by Mplus 8.3.

Reliability was assessed using Cronbach’s α and test–retest reliability to evaluate the internal consistency of the items, while Pearson’s correlation coefficient was used to examine the stability between the two measurements.

For validity, three psychology experts were consulted for content evaluation. Confirmatory factor analysis (CFA) was conducted to assess model fit using indices such as CFI, TLI, and RMSEA. Criterion-related validity was examined through correlation analysis, testing the relationships between the self-love scale and general self-confidence, self-reliance, and health-promoting lifestyles.

### 3.3. Results

**Reliability:** Cronbach’s α coefficient: After analyzing the reliability of 22 items, the results showed that Cronbach’s α coefficient was 0.78 (*M* = 4.02, *Min* = 0.30, *Max* = 0.30).

Internal consistency: Each indicator showed a significant correlation with the total score, ranging from 0.40 to 0.70 (Table 5).

Test–retest reliability: After six weeks, the test–retest reliability was assessed using the correlation coefficient between the two measurement points from the 163 collected samples. The overall correlation was 0.66, indicating the moderate stability of the scale over time.

**Validity:** The validity test mainly included content validity, confirmatory factor analysis, and criterion-related validity.

Content validity: The items and indicators of the scale were constructed on the basis of a literature review and qualitative research. Three psychology professors evaluated whether each item belonged to self-love and its sub-indicators; the results showed that the items were more in line with the five indicators.

Confirmatory factor analysis of data from 929 Chinese adults indicated a strong fit for the five-factor model. According to the fitness index and the evaluation standard of the structural equation modeling (the relationship between latent variables) of the overall model, it was found that the absolute fitness index was the best in this study; the χ^2^ (hypothesis test through Chi-square) was 544.95, and the root mean square error of approximation (RMSEA) was 0.048, which was less than 0.05, indicating that the fit was good; in the value-added fitness index, the values of the comparative fit index (CFI) and Tucker–Lewis index (TLI) were above 0.90; the simple fitness index demonstrated that *χ*^2^/*df* = 2.73, that is, 1 < *χ*^2^/*df* < 3, which indicated that the model displayed parsimony fitness (Table 6). The details of the path analysis of self-love can be seen in Figure 1.

Criterion-related validity: The validity of the questionnaire needed to be investigated in many respects, not only through structural validity. Hence, we also used the criterion to conduct the criterion-related validity, which aided in verifying the validity of the questionnaire.

Partial correlation between self-love and the related Chinese Adult SLS: After controlling for variables such as age, gender, registered residence, marital status, and family income, we found that self-love was significantly correlated with [5] ([5]); they studied participants’ feelings of love and [2]’s ([2]) SASB scale. However, there was no correlation between self-acceptance and self-love in either the least optimal state or in the degree of feeling loved (Table 7).

**Partial correlation between self-love and general self-confidence**: All indicators other than self-respect correlated with general self-confidence (Table 8).

**Partial correlation between self-love and sense of responsibility:** The partial correlation results between self-love and responsibility demonstrated that there was a significant correlation between overall self-love, self-persistence, self-restraint, and responsibility; however, there was no correlation between self-respect and responsibility (Table 9).

**Partial correlation between self-love and health-promoting lifestyles**: The relationships between self-love and its indicators as well as the dimensions of a health-promoting lifestyle are displayed in Table 8. The overall self-love score was positively correlated with health responsibility, interpersonal relationships, stress management, nutrition, spiritual growth, physical activity, and overall health promotion. Self-respect, self-acceptance, and a health-promoting lifestyle were not related (Table 10).

### 3.4. Discussion

**Reliability:** In terms of reliability, the α coefficient was the benchmark for the reliability coefficient, which is accepted by the psychology and psychometrics community ([34]).

The reliability analysis demonstrated that the scale possessed acceptable internal consistency, as indicated by the Cronbach’s α coefficient. This suggested that the items consistently measured the intended construct and that the scale was suitable for research purposes. Internal consistency analysis further confirmed that each item was significantly correlated with the total score, indicating that all items contributed meaningfully to the overall construct.

These findings support the scale’s reliability and suggest that it effectively captured the underlying concepts. However, future research could further examine the scale’s structure to identify potential areas for refinement. Exploring item performance across different samples and contexts may provide additional insights into its stability and applicability ([25]).

**Validity:** The items of this scale were derived from a corpus of ancient books and the content of interviews with modern adults, which is why its theoretical foundation was solid. On this basis, three psychological experts were invited to evaluate the content validity.

The RMSEA results indicated a good model fit. Additionally, among the value-added fit indices, both CFI and TLI met the established psychometric standards. The structural model of the five factors in the results of this study was in line with previous theoretical assumptions and was apt for the model of this scale ([32]).

In this study, age, gender, registered residence, and children were all controlled for. To a certain extent, the influence of demographic variables was controlled for, and the relationship between self-love and other criteria could be better explored.

## 4. General Discussion

This study developed and validated the Chinese Adult Self-Love Scale (SLS), identifying five indicators: self-cherishing, self-acceptance, self-restraint, self-kindness, and self-discipline. These indicators were derived from qualitative interviews and supplemented with content from *Siku Quanshu*. Through a series of reliability and validity assessments, we established a 22-item scale grounded in five core indicators. This scale provided a robust foundation for understanding how self-love is structured and expressed within the Chinese cultural context, while also offering a culturally grounded reference point for future comparisons with self-love as conceptualized in Western psychological traditions.

### 4.1. The Inheritance and Development of Self-Love

The concept of self-love has been deeply embedded in Chinese philosophy since ancient times and has continuously evolved with societal changes. In ancient China, self-love was primarily socially oriented, emphasizing the “greater self” over individual needs. This understanding was shaped by strict legal punishments, fostering a model of self-restraint and moral discipline rather than prioritizing personal well-being ([13]). Confucianism, for example, positioned self-love within the framework of responsibility, propriety, and societal harmony, reinforcing the idea that loving oneself meant adhering to moral obligations and maintaining social order. The findings of this study reaffirm these traditional values, identifying self-cherishing, self-acceptance, and self-restraint as core indicators of self-love.

However, while traditional perspectives often framed self-love within social expectations and collective obligations, modern society’s understanding of self-love is gradually shifting toward a balance between personal well-being and social responsibility. [16] ([16]) noted that cultural transformation has been accompanied by a rise in individualism and a decline in collectivism, which has influenced how self-love is perceived and practiced. This study highlights an evolving perspective that integrates both self-discipline and self-care, reflecting a broader societal transition in contemporary Chinese culture.

Nevertheless, despite this gradual shift toward individual well-being, self-love in China has not completely aligned with Western individualism. While the emphasis on psychological well-being and self-compassion has grown, self-love in the Chinese context still retains a social dimension, emphasizing collective harmony and moral responsibility. This distinction becomes more apparent in cross-cultural comparisons.

### 4.2. Cross-Cultural Differences and Commonalities

While cross-cultural dialogues about self-love have long existed, effective measurement tools can enhance and refine these discussions by providing structured insights. Western perspectives on self-love predominantly focus on self-esteem, self-assertion, and personal autonomy. For instance, [7] ([7]) and [8] ([8]) conceptualized self-love as either high self-esteem or narcissism, emphasizing its role in individual identity rather than social responsibility. Similarly, [2] ([2]) highlighted the relationship between self-love and mental health, arguing that self-love serves as the foundation of personal happiness.

In contrast, Chinese self-love retains a collective component, integrating family, societal expectations, and national identity. This distinction is reflected in the inclusion of “I love my nation and country” as a key item in the scale, underscoring the idea that self-love is not solely about individual self-care but also about fulfilling social and moral responsibilities. The Chinese notion of self-love thus embodies both the “small self” (individual self-love) and the “greater self” (collective self-love) ([39]). While self-love in China is evolving toward a more personal orientation, it remains fundamentally tied to interpersonal relationships and social harmony, making it distinct from Western notions that prioritize individual fulfillment above communal well-being.

Therefore, future research can further explore how self-love in China continues to evolve amidst shifting cultural values. Expanding cross-cultural comparisons may help to elucidate whether self-love in China is moving toward a more Western-style individualism or whether it will maintain its unique balance between self-fulfillment and social obligation.

## 5. Limitation and Prospects

Despite its contributions, this study has several limitations that should be acknowledged. Addressing these limitations in future research will further strengthen the understanding of self-love in cultural contexts.

First, the study relied entirely on self-reported measures, which may be subject to social desirability bias. Participants might have responded in ways that aligned with societal expectations rather than their genuine self-perceptions. Future studies could incorporate multi-method approaches, such as behavioral assessments or peer reports, to improve data validity.

Second, data collection took place during the COVID-19 pandemic, a period that may have influenced participants’ emotional states and perceptions of self-love. Longitudinal studies or post-pandemic replications are recommended to examine the stability of and changes in self-love over time.

Third, this study did not compare self-love with other closely related self-concepts such as self-esteem, self-compassion, and narcissism within a cultural framework. Future research can build upon the current scale to conduct cross-construct and cross-cultural comparisons, and to explore the role of self-love in psychological well-being, clinical interventions, and personal development.

In sum, this study offers a culturally informed and psychometrically sound instrument for assessing self-love among Chinese adults. It lays a solid foundation for future research and contributes to a deeper understanding of the complex, culturally embedded nature of self-love.

## 6. Conclusions

The results of this study demonstrate that Chinese adults’ self-love can be ultimately determined using 22 items, including five indicators: self-cherishing, self-acceptance, self-restraint, self-kindness, and self-discipline. After conducting the reliability and validity tests, we established the α coefficient and test–retest reliability, content validity, structural validity, and criterion-related validity.

## Figures and Tables

**Figure 1 behavsci-15-00652-f001:**
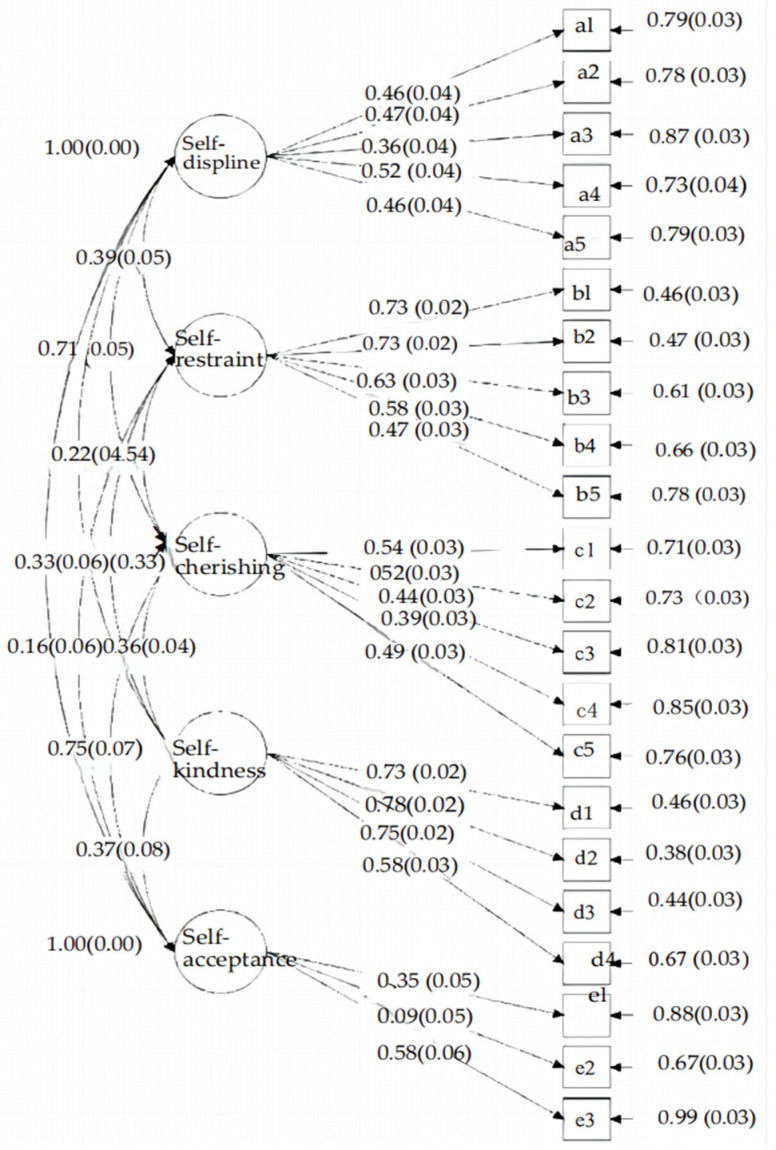
The path analysis diagram of self-love and its indicators (*n* = 929).

**Table 1 behavsci-15-00652-t001:** Hypotheses of the Chinese Adult Self-Love Scale (SLS).

Hypothesis	Theoretical Basis (*Siku Quanshu*)	Supporting Quotes ([17])	Theoretical Basis (Interviews with Contemporary Individuals)	Supporting Quotes
**Hypothesis 1:** *Self-love includes at least the indicator of self-cherishing.*	The ancient people cared for and cherished themselves and everything related to their existence, such as life, body, property, future, and status.	The people who did not live up to their anticipated life expectancy or suffered from premature deaths were those who showed no caring toward themselves. (Vol. 32, Seven Lots from the Bookbag of the Clouds)	The modern adults cherished, protected, cared for, and respected oneself and others	Of course, there are physical issues to consider… If I love myself, then I am physically and mentally fine—healthy and balanced (z003).
**Hypothesis 2:** *Self-love includes at least the indicator of self-acceptance.*	The ancients were able to accept themselves, and their appreciation was regardless of their circumstances.	Anyone who hears these words cannot help but feel awakened, gaining a sense of clarity and transcendence, and learning to cherish themselves (Vol. 9, Chen Yazhi’s Poetry)	The modern adults could accept one’s current reality and develop and pursue an ideal self.	The first step in self-love is to know yourself and embrace your individuality… Because this is important, because you were born this way, you don’t get to decide what you’re born with. Can’t decide (z018).
**Hypothesis 3:** *Self-love includes at least the indicator of self-restraint.*	The ancients upheld laws, morals, and order, practicing moderation and righteousness while avoiding recklessness. The state enforced strict control through harsh corporal punishment.	In the past, people valued self-respect and were cautious about breaking the law. But today, with punishments reduced to mere flogging, it is no surprise that the sense of integrity and shame has been lost (Vol. 49, Inscriptions on Stone)	Modern adults follow basic societal norms, behaving with caution and propriety, and make self-discipline and careful solitude central to their way of living.	I believe self-restraint requires a sense of caution and independence… This kind of faith, perhaps what sustains your aboveboard behavior is more based on self-love, and what sustains order is also based on self-love (z007).
**Hypothesis 4:** *Self-love includes at least the indicator of self-responsibility.*			Modern adults believe that individuals who practice self-love should possess a sense of responsibility.	I can feel that she is very responsible to her family…who have some relationship with her, for some of their life, career guidance (z006).
**Hypothesis 5:** *Self-love includes at least the indicator of Self-respect.*			Modern adults believe that self-love encompasses having one’s own opinions, principles, boundaries, limits, and dignity in both actions and interactions.	I feel that there is a certain bottom line; that is, when pursuing one’s dream, one should not exceed one’s bottom line just because of some interests (z017).

**Table 2 behavsci-15-00652-t002:** Evaluation of the items of Chinese Adult SLS.

Evaluation Process	Evaluation Content	Example	Evaluation Criterion	Evaluation Time
First round	Relevance degree	Please rate this according to the relevance between the following sentences and “cherish yourself”	The evaluation scale ranged from 1 (very irrelevant) to 5 (highly relevant).	22–27 December 2019
	Language clarity	I cherish my health; I cherish my life; I cherish my family; and so on	The evaluation scale ranged from “very clear”, “clear”, “general”, and “fuzzy”, to “very fuzzy”.	22–27 December 2019
Second round	Language clarity	Same as above	Same as above	5–7 January 2020

**Table 3 behavsci-15-00652-t003:** Evaluation of the items of the Chinese Adult SLS.

Items	First Round *M*	Second Round *M*	Items	First Round *M*	Second Round *M*
I am strict with myself.	5.0	4.64	I can accept and allow myself to make mistakes.	4.9	4.55
I am a self-disciplined person.	5.0	4.45	I can face my true self with ease.	5.0	4.55
My life is very regular.	4.8	4.45	I never ask too much of myself.	4.9	4.36
I think desire is inevitable.	4.6	4.18	I strive to reach my full potential.	4.9	4.64
In life, I can set an example for others using my behavior.	5.0	—	I am working towards becoming the person I want to be.	4.8	4.64
I abide by the law.	5.0	5.0	I never stop learning.	4.6	4.36
I never indulge in self-degradation.	4.9	4.36	I am continuously improving myself.	4.6	4.36
I observe the social order.	5.0	4.91	My personal needs are reasonable and moderate.	4.6	4.18
I love my country and nation.	4.6	4.91	I never indulge in self-degradation.	4.9	4.36
I never do anything against my morality.	4.8	4.27	I think desire is inevitable.	4.6	4.18
I insist on the integrity of justice and action.	5.0	4.55	I think I am more important than others.	4.9	4.36
I cherish my future very much.	4.8	4.27	I want to be who I really want to love.	4.8	4.64
I cherish my life very much.	5.0	4.73	I take responsibility for my actions.	3.9	—
I cherish my abilities.	4.9	4.55	I know what I should and should not do.	4.3	—
I don’t begrudge my abilities for the sake of my family and society.	4.5	4.55	In life, I fulfill my roles properly.	4.5	—
I do my best to protect myself.	4.9	4.27	I set an example for others through my actions.	5.0	—
I never force myself to do what I don’ t want to do.	4.1	4.55	I cherish and care for my family.	4.8	—
In life, I never hurt myself.	4.9	4.36	I respect myself and earn the respect of others.	4.9	4.18
In life, I seldom embarrass myself.	4.6	4.18	I have a clear understanding of myself.	4.6	4.27

**Table 4 behavsci-15-00652-t004:** Items and indicators retained after exploratory factor analysis.

Items	Self-Discipline	Self-Restraint	Self-Cherishing	Self-Kindness	Self-Acceptance	*M*	*SD*
1. I am strict with myself.	0.82					3.51	0.88
2. I am a self-disciplined person.	0.81					3.45	0.97
3. My life is very regular.	0.69					3.34	1.00
4. I never stop learning.	0.65					3.43	0.93
5. In life, I can set an example for others using my behavior.	0.61					3.70	0.76
6. I abide by the law.		0.79				4.56	0.70
7. I observe the social order.		0.76				4.47	0.70
8. I love my country and nation.		0.73				4.64	0.68
9. I never do anything against my morality.		0.65				4.27	0.77
10. I insist on the integrity of justice and action.		0.63				4.21	0.78
11. I cherish my future very much.			0.72			4.16	0.80
12. I cherish my life very much.			0.67			4.27	0.81
13. I cherish my abilities.			0.66			4.00	0.75
14. I don’t begrudge my abilities for the sake of my family and society.			0.62			4.01	0.80
15. I do my best to protect myself.			0.61			4.17	0.73
16. I never force myself to do what I don’t want to do.				0.74		3.42	0.96
17. In life, I never hurt myself.				0.73		3.27	0.90
18. In life, I seldom embarrass myself.				0.72		3.47	0.86
19. I never ask too much of myself.				0.71		3.47	0.89
20. I think desire is inevitable.					0.79	4.10	0.80
21. I think I am more important than others.					0.76	3.35	1.00
22. I want to be who I really want to love.					0.60	4.18	0.79

**Table 5 behavsci-15-00652-t005:** Internal consistency (*n* = 929).

Indicators	Total Self-Love
Self-cherishing	0.70
Self-acceptance	0.40
Self-restraint	0.57
Self-discipline	0.70
Self-kindness	0.68

**Table 6 behavsci-15-00652-t006:** Results of the confirmatory factor analysis (*n* = 929).

Model	*χ* ^2^	*df*	*χ*^2^/*df*	RMSEA	CFI	TLI
Five-indicator model	544.95	199	2.73	0.048	0.91	0.90

**Table 7 behavsci-15-00652-t007:** Partial correlation between self-love and indicators in different states (*n* = 364).

Variable	Self-Discipline	Self-Restraint	Self-Cherishing	Self-Kindness	Self-Acceptance	Self-Love
Love of self	0.30	0.17	0.42	0.26	0.27	0.46
Feelings of love	0.29	0.25	0.25	0.16	0.08	0.35
Self-love in its most optimal state	0.27	0.20	0.41	0.18	0.21	0.40
Self-love at its least optimal state	0.33	0.15	0.26	0.19	0.06	0.35

**Table 8 behavsci-15-00652-t008:** Partial correlation between self-love and general self-confidence (*n* = 286).

Variable	Self-Discipline	Self-Cherishing	Self-Kindness	Self-Acceptance	Self-Restraint	Self-Love
General self-confidence	0.67	0.44	0.13	0.37	0.47	0.67

**Table 9 behavsci-15-00652-t009:** Partial correlation between self-love and responsibility (*n* = 374).

Variable	Self-Discipline	Self-Restraint	Self-Cherishing	Self-Kindness	Self-Acceptance	Self-Love
Responsibility	0.45	0.39	0.37	0.14	0.17	0.48

**Table 10 behavsci-15-00652-t010:** Partial correlation between self-love and health-promoting lifestyles (*n* = 163).

Variable	Self-Discipline	Self-Restraint	Self-Cherishing	Self-Kindness	Self-Acceptance	Self-Love
Health responsibility	0.33	0.17	0.13	0.11	0.01	0.32
Relationships	0.22	0.33	0.36	0.11	0.18	0.33
Pressure management	0.19	0.23	0.22	0.03	0.20	0.25
Nutrition	0.38	0.21	0.20	0.12	0.07	0.37
Spiritual growth	0.31	0.29	0.29	0.06	0.20	0.38
Physical activity	0.27	0.11	0.10	0.11	0.04	0.23
Health-promoting lifestyle	0.40	0.29	0.27	0.13	0.16	0.43

## Data Availability

The datasets generated during and/or analyzed during the current study are available from the author upon reasonable request.

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
