# Peer review of "Inheritance and Development of Self-Love: A Measurement Based on Chinese Adults"

_behavsci, 2025, doi:10.3390/bs15050652_

Round 1
Reviewer 1 Report
Comments and Suggestions for Authors
The manuscript is relevant to the topic addressed and appropriate for the journal. It clearly follows the general steps of validating a psychological instrument. In general, the text is clear, easy to follow and to understand, with some exceptions to be notified below. The authors need to revise certain statements, include observations related to the statistical results, and work further on the arguments of the paper and the implications of the results for research.
- In the introduction the literature is presented less analytical and critical way.
- I suggest you present what scales for measuring self-love exist in the literature. Make a brief inventory.
- The motives of the study are not convincing enough to construct a psychological instrument.
- There are exaggerated statements: partially complete, or simplified statements of the ideas of the cited authors.
Rows 38 -39, "most psychologists..." and only two researchers are given in parentheses- Leary & Tangney, 2012; this reference is not listed in the bibliography.
-There are partially complete or simplified statements of the ideas of the cited authors.
For example: Rows 41-42: "Their research indicates that there are no significant differences in these emotional experiences between Chinese and American university students (Brown, 2007; Cai et al., 2007)" .
But Cai et al. 2007, show that cultural differences in global self-esteem reflect cognitive rather than affective factors. "Study 2 found that cultural differences in modesty underlie cultural differences in cognitive self-evaluations. These findings suggest that Chinese feel as positively toward themselves as Americans do, but are less inclined to evaluate themselves in an excessively positive manner."
-Rows 52- 53 must be rephrased. Philosophical conceptions of self-love do not need to have empirical validation (to be valuable), but they can inspire the definition of a psychological construct.
-In Rows 54-56 - The accuracy of the quoted ideas is not respected: "Although self-love is important to human civilization, it has not been studied in depth within the field of psychology because most researchers consider it to be a basic, common, and implicit motivation (Brown, 2010; Kitayama & Uchida, 2003; Kobayashi & Greenwald, 2003; Yamaguchi et al., 2007).
For example Brown (2010) summarizes that: "Moreover, in both cultures (and to a largely comparable degree), individual differences in self-esteem predict (d) psychological well-being and (e) emotional regulation in response to negative outcomes. These commonalities suggest that self-love is a universal human motivation".
Comparing the two texts above, it can be seen that the meaning is different between how the authors of the manuscript interpret the ideas of the cited author.
-Row 68 - What are effective measurement tools? In psychology we work with scientifically validated tools.
-Rows 69 -74 - A valid psychological tool need not necessarily be derived from a deep culture. In this sense, Cai's scale is no less valid. Reformulate!
-Rows 82-86 give the operational definition of self-love and row 88 asks: "However, can this concept be verified?"
The stake of the study seems to be the empirical verification of philosophical concepts of self-love. Confusingly, it seems that the aim of the study indicated in the introduction and in R 93-96 is not consistently pursued.
-Rows 90-91- "Therefore, to understand the background of Chinese culture based on the existing qualitative research, we developed the Chinese Adult Self-Love Scale (SLS)..."
Was the instrument constructed to understand the background of Chinese culture? This purpose can be achieved without constructing a psychological instrument. Reformulate!
-R102- Not clear. Is the interview with Chinese adults conducted by the authors authors of the manuscript or by Xue et al., 2021; Xue et al., 2020?
-In 2.2 it is not stated how many items were originally, how many were deleted and why? These data after the first round of pretesting should be indicated in the table in detail. 68 items remained from how many?
-At 2.5 There were 22 items remaining grouped into 5 indicators (factors) after exploratory factor analysis. Present as figure the proportion of variance by factor exported from SPSS.
-To Table 3 add to the table the mean (M) and standard deviation (SD) for each item.
-R179-181 - How do you explain this combination of factors (3 from traditional Chinese culture and 2 from modern Chinese culture)?
-R193-194 -Which authors? Give references.
-3.2 State the Klenbach's alpha for each of the 5 indicators with export evidence from SPSS. It is said to be 0.77 for each of them. Evidence needed.
-Do not specify in the tables the significance of *, **, ***.
-R355-356 – There is not Wang, 2014 in references.
- R364-365 - It is unclear. Reword.
-Subchapter 4 "General Discussion and Prospects" concludes the study data. What is the point if there is also subchapter 5 Conclusions?
-4.2 can be put in the introduction when discussing the richness of self-love in Chinese cultural texts or you can argue what its role is here. It is not clear to the reader.
-R426-427 - " Only with effective measurement tools can one derive an effective dialog with self-love in other cultures as well." Cross-cultural dialogues about self-love have long existed without self-love measurement tools. Restate.
-R427-435 - A comparison of the conceptualizations of self-love in Western and Chinese psychological research is attempted but it is uncritical, without bringing new observations.
-It does not convincingly argue what this tool will be used for, to whom, scopes and other benefits for Chinese and Western psychological research.
-What are the limitations of the study? They are not specified.
Comments on the Quality of English LanguageNot the case.
Author Response
Dear Editor and Reviewers,
Thank you for your valuable feedback on our manuscript. In response to the reviewers' comments, we have made comprehensive revisions. Notably, we have reconsidered the naming of key indicators, now referred to as self-kindness and self-discipline, and have thoroughly refined the content of Study 1 and Study 2. Please refer to the revised manuscript for further details.
Best regards,
Liming Xue
Reply Review1
First and foremost, I sincerely appreciate your meticulous and detailed feedback on this manuscript, as well as your insightful interpretation of Professor Cai’s work. Your comments have made me realize my misunderstanding of the relationship between self-love and self-esteem in this section.
Below, I have made revisions based on your suggestions. Due to textual adjustments, the line numbers may have shifted slightly. To ensure clarity, I have highlighted the modified sections in **blue** and included them in my response.
Once again, thank you for your valuable time and thoughtful guidance.
Open Review1
The manuscript is relevant to the topic addressed and appropriate for the journal. It clearly follows the general steps of validating a psychological instrument. In general, the text is clear, easy to follow and to understand, with some exceptions to be notified below. The authors need to revise certain statements, include observations related to the statistical results, and work further on the arguments of the paper and the implications of the results for research.
- In the introduction the literature is presented less analytical and critical way.
- I suggest you present what scales for measuring self-love exist in the literature. Make a brief inventory.
- The motives of the study are not convincing enough to construct a psychological instrument.
- There are exaggerated statements: partially complete, or simplified statements of the ideas of the cited authors.
Comments 1:-Rows 38 -39, "most psychologists..." and only two researchers are given in parentheses- Leary & Tangney, 2012; this reference is not listed in the bibliography.
Reply 1:Thank you for pointing this out. I add this citation in the bibliography in R497
Comments 2:-There are partially complete or simplified statements of the ideas of the cited authors.
For example: - Rows 41-42: "Their research indicates that there are no significant differences in these emotional experiences between Chinese and American university students (Brown, 2007; Cai et al., 2007)" .
But Cai et al. 2007, show that cultural differences in global self-esteem reflect cognitive rather than affective factors. "Study 2 found that cultural differences in modesty underlie cultural differences in cognitive self-evaluations. These findings suggest that Chinese feel as positively toward themselves as Americans do, but are less inclined to evaluate themselves in an excessively positive manner."
Reply 2:Thank you for pointing this, I rewrote this in Rows 47-49: Cai et al. (2007) have used four self-relevant emotions (ashamed, humiliated, proud, pleased), they suggest that self-love was equally strong in East Asia as in North America and Western Europe.
Comments 3:-Rows 52-53 must be rephrased. Philosophical conceptions of self-love do not need to have empirical validation (to be valuable), but they can inspire the definition of a psychological construct.
Reply 3:Thank you for pointing this, and it give me a new sight. So I changed this: “Philosophical perspectives on self-love, while not requiring empirical validation, offer valuable insights that can inspire the development of psychological constructs”in Rows 34-35.
Comments 4:-In Rows 54-56 - The accuracy of the quoted ideas is not respected: "Although self-love is important to human civilization, it has not been studied in depth within the field of psychology because most researchers consider it to be a basic, common, and implicit motivation (Brown, 2010; Kitayama & Uchida, 2003; Kobayashi & Greenwald, 2003; Yamaguchi et al., 2007).
For example Brown (2010) summarizes that: "Moreover, in both cultures (and to a largely comparable degree), individual differences in self-esteem predict (d) psychological well-being and (e) emotional regulation in response to negative outcomes. These commonalities suggest that self-love is a universal human motivation".
Comparing the two texts above, it can be seen that the meaning is different between how the authors of the manuscript interpret the ideas of the cited author.
Reply 4:Thank you for pointing this out. After carefully rereading the original source, I recognize that my previous interpretation of Brown (2010) was inaccurate. I revised in Rows 44-57:
Although self-love is deeply rooted in philosophical traditions, most psychological studies conceptualize it as basic, common, and implicit motivation (Brown, 2010; Leary & Tangney, 2012; Kitayama & Uchida, 2003; Kobayashi & Greenwald, 2003; Yamaguchi et al., 2007). Researchers such as Cai et al. (2007) have used four self-relevant emotions (ashamed, humiliated, proud, pleased), they suggest that self-love was equally strong in East Asia as in North America and Western Europe. While Bruce et al. (2019) employed the questions 'Do you love yourself?' and 'How much do you love yourself?' to measure self-love, they found there was significant difference between races(Causasian, Non-Caucasian and more than one race). The inconsistencies between the above two findings may reflect cultural and ethnic differences in measurement tools, and also challenge the conclusion that self-love is universal or culturally specific. Therefore, we need measurement tools that can take into account cultural contexts and be adapted to different social contexts to more accurately capture the multidimensional characteristics and cultural adaptation of self-love.
Comments 5:-Row 68 - What are effective measurement tools? In psychology we work with scientifically validated tools.
Reply 5:Thank you for your insightful perspective; it has given me valuable inspiration. However, my viewpoint is as follows: since this study focuses on scale development from a psychometric perspective, I place greater emphasis on the standardized process of scale construction. In my view, conducting reliability and validity testing is a necessary criterion for determining the effectiveness of a standardized measurement tool.
Comments 6:-Rows 69 -74 - A valid psychological tool need not necessarily be derived from a deep culture. In this sense, Cai's scale is no less valid. Reformulate!
Reply 6:Thank you for pointing this out. I revised in reply 4.
Comments 7: -Rows 82-86 give the operational definition of self-love and row 88 asks: "However, can this concept be verified?"
The stake of the study seems to be the empirical verification of philosophical concepts of self-love. Confusingly, it seems that the aim of the study indicated in the introduction and in R 93-96 is not consistently pursued.
Reply 7:Thank you for pointing this out. I revised the aims of studies in R70-74: “To bridge the gap between philosophical perspectives and psychological conceptualization, we aim to empirically validate a culturally grounded framework of self-love and operationalize it as a measurable psychological construct. This study seeks to determine whether the proposed scale accurately reflects the lived experiences of individuals in China and how self-love manifests differently across cultural contexts.
Comments 8:-Rows 90-91- "Therefore, to understand the background of Chinese culture based on the existing qualitative research, we developed the Chinese Adult Self-Love Scale (SLS)..."
Was the instrument constructed to understand the background of Chinese culture? This purpose can be achieved without constructing a psychological instrument. Reformulate!
Reply 8:To bridge the gap between existed qualitative researches and psychological conceptualization, we aim to empirically validate a culturally grounded framework of self-love and operationalize it as a measurable psychological construct. We developed Chinese self-love scale(SLS) to measure self-love reflects the lived experiences of individuals in China and how self-love manifests differently across cultural contexts.
Comments 9: -R102- Not clear. Is the interview with Chinese adults conducted by the authors authors of the manuscript or by Xue et al., 2021; Xue et al., 2020?
Reply 9:Thank you for pointing this out. I revised rows Compare that to this, within the Confucian-influenced Chinese cultural context, our previous content analysis of SiKu QuanShu...in Rows 58-59 and64-65.
Comments 10: -In 2.2 it is not stated how many items were originally, how many were deleted and why? These data after the first round of pretesting should be indicated in the table in detail. 68 items remained from how many?
Reply 10:Thank you for pointing this out, I write the originally items counts in Rows105-106:(1)“According to the guidelines of Dewelis (2016), 90 items were initially prepared with certain redundancy”(Rows91-92). (2) 68 items remained from 90 items.
Comments 11: -At 2.5 There were 22 items remaining grouped into 5 indicators (factors) after exploratory factor analysis. Present as figure the proportion of variance by factor exported from SPSS.
Reply 11: The respective percentages of the five factors are: 27.90%, 10.20%, 8.68%, 6.09% and 4.80%, they cumulatively accounted for 58.4% of the variance.
Comments 12: -To Table 3 add to the table the mean (M) and standard deviation (SD) for each item.
Reply 12:Thank you for pointing this out, I add the mean (M) and standard deviation (SD) in table 4.
Comments 13:-R179-181 - How do you explain this combination of factors (3 from traditional Chinese culture and 2 from modern Chinese culture)?
Reply 12:Thank you for pointing that, I rewrote the discussion in Rows 190-220.
Comments 14:-R193-194 -Which authors? Give references.
Row180:(Xue, 2021; Xue et al., 2020).
Comments 15:-3.2 State the Klenbach's(Cronbach's) alpha for each of the 22items with export evidence from SPSS. It is said to be 0.77 for each of them. Evidence needed.
Reply 15:After analyzing the reliability of 22 items, the results showed that Cronbach's α coefficient was 0.78(M=4.02, Min=0.30, Max=0.30).
Comments 16:-Do not specify in the tables the significance of *, **, ***.
Reply 16: Thank you for pointing this out, I revised them.
Comments 17:-R355-356 – There is not Wang, 2014 in references.
Reply 17: Thank you for pointing this out, I added it in R540-541. Wang, K. (2012). Ren zhe zi ai: Rujia chuantong de daode shengming guan ji qi zhexue jichu. Kongzi Yanjiu, (5), 22-31.
Comments 18:- R364-365 - It is unclear. Reword.
Reply 18:Thank you for pointing this out, I revised R363-365. The RMSEA results indicated a good model fit. Additionally, among the value-added fit indices, both CFI and TLI met the established psychometric standards.
Comments 19:-Subchapter 4 "General Discussion and Prospects" concludes the study data. What is the point if there is also subchapter 5 Conclusions?
Reply 19:Thank you for pointing this out, I rewrote the part of general discussion and conclusion.
Comments 20:-4.2 can be put in the introduction when discussing the richness of self-love in Chinese cultural texts or you can argue what its role is here. It is not clear to the reader.
Reply 20:Thank you for pointing this out, I rewrote introduction.
Comments 21:-R426-427 - " Only with effective measurement tools can one derive an effective dialog with self-love in other cultures as well." Cross-cultural dialogues about self-love have long existed without self-love measurement tools. Restate.
Reply 21:Thank you for pointing this out, I revised the sentence. While cross-cultural dialogues about self-love have long existed, effective measurement tools can enhance and refine these discussions by providing structured insights.
Comments 22::-R427-435 - A comparison of the conceptualizations of self-love in Western and Chinese psychological research is attempted but it is uncritical, without bringing new observations.
Comments 23:-It does not convincingly argue what this tool will be used for, to whom, scopes and other benefits for Chinese and Western psychological research.
Reply 23:The scale is primarily designed for adults aged 18 and above, with a certain level of education, within the context of Chinese culture. It can also be applied in value education and psychological interventions within this framework. Furthermore, the scale has cross-cultural applicability, particularly in countries with Confucian cultural backgrounds, and can serve as an important tool for cultural comparison in Western psychological research on self-love.
Comments 24:-What are the limitations of the study? They are not specified.
Reply 24:The limitations of this study are as follows:
- Self-report bias: Participants may be influenced by social desirability bias, providing responses that align with societal norms rather than their true self-assessment.
- Impact of COVID-19: The pandemic may have affected self-love perceptions and emotions. Future studies could explore post-pandemic changes or conduct longitudinal research on its long-term impact.
- Cultural diversity in China: Cultural differences between generations and regions within China may influence self-love perceptions. Future research should consider these generational and social group differences.
Reviewer 2 Report
Comments and Suggestions for Authors
The study attempts to develop a self-love measurement scale specific to Chinese cultural values, filling a gap in cross-cultural psychological assessments.
While it argues for the uniqueness of self-love in Chinese culture, it does not provide a sufficiently strong argument for why existing self-esteem or self-love scales (from Western research) are inadequate. The study primarily builds on prior qualitative research, but it does not propose a fundamentally novel theoretical framework.
While the research is important for Chinese psychology, it lacks a discussion on how the findings might be generalized or adapted for non-Chinese populations. Given that self-love is a universal human experience, failing to contextualize it within broader cross-cultural perspectives weakens its applicability.
The objectives do not explicitly mention how this scale will be applied in real world.
The study follows a standard scale development process. But needs to acknowledge in the following bias (add a Limitation section into the Discussion):
Sampling bias: The study relies on snowball sampling through WeChat, which may introduce selection bias, limiting generalizability.
Ethical approval vagueness: The manuscript states IRB approval but does not specify the reference number or detailed ethical considerations.
The process of item refinement lacks justification for why certain items were retained or discarded.
It needs more justification on the five-factor model. Why were these five factors chosen, and how do they compare with prior self-love/self-esteem models?
Lack of cross-validation: The study does not include a test-retest reliability analysis, which raises concerns about the longitudinal stability of the scale.
Insufficient external validation: The study compares its scale with some Western self-love/self-esteem scales, but it does not include real-world behavioral validation. For example, how does the scale correlate with actual self-care behaviors, mental health outcomes, or resilience?
The practical implications are weakly discussed. How can this scale be applied in mental health assessments, therapy, or educational settings?
It needs to address potential limitations in generalizing results across different Chinese subcultures (e.g., urban vs. rural, generational differences).
Grammatical errors and awkward phrasing appear throughout. "Finally, we discussed the inheritance and development of self-love among Chinese people." → “We then analyzed how self-love is inherited and developed among Chinese adults.”
Inconsistent use of tenses (e.g., past vs. present).
Overuse of direct translations from Chinese concepts without proper contextualization for an international audience.
The discussion section repeats many points from the results section.
The introduction is too lengthy, with excessive theoretical background.
The results and discussion sections blend together, making it difficult to distinguish raw data from interpretation.
The tables are not well explained. Some factor loadings could be visualized better (e.g., using a path diagram for the factor structure).
Line-by-Line Comments
Line 9. "Self-love is a prominent and preeminent quality..." => Redundant phrasing. Use either "prominent" or "preeminent."
Line 18. “The Klenbach’s α..." => It should be "Cronbach’s α". Please revise it throughout the manuscript.
Line 19. "The confirmatory factor indicators were above 0.90." =>Specify which indicators (CFI, TLI, RMSEA).
Line 39. "They equate self-love with self-esteem..." => False generalization. Some psychologists differentiate self-love and self-esteem.
Line 79. Please explain the concept “Sikuquanshu” and add citations.
Line 105. Table 1. is unclear in explicitly linking each hypothesis to its corresponding theoretical basis. The hypotheses are not explicitly numbered or visually aligned with their theoretical bases. The theoretical basis is provided as a block of text without clear demarcations. The relationship between hypothesis and example citations is implied rather than explicitly stated. Please use the format:
Hypothesis |
Theory Basis |
Supporting Quotes |
H1. …. |
Sikuquanshu, lack of self-care leads to premature death (Vol. 32). |
“If I love myself, then I am physically and mentally fine—healthy and balanced” (Interview z003). |
H2… |
|
|
Line 132. "The participants could transmit the questionnaire and receive extra compensation." => Clarify: How to get compensation? Could this bias responses?
Line 170. "SPSS was used ..." => Mention version number of SPSS (e.g., SPSS 26.0).
Several references are incomplete (e.g., missing DOI, page numbers).
APA format => needs to change to the Journal format.
Author Response
Author:Thank you very much for this valuable opportunity to revise my manuscript. I have carefully addressed each of your comments and provided detailed responses, which I have highlighted in blue. Additionally, based on your suggestions, I have further refined and reorganized the manuscript to enhance its clarity and coherence.Notably, we have reconsidered the naming of key indicators, now referred to as self-kindness and self-discipline, and have thoroughly refined the content of Study 1 and Study 2. Please refer to the revised manuscript for further details.
The study attempts to develop a self-love measurement scale specific to Chinese cultural values, filling a gap in cross-cultural psychological assessments.
While it argues for the uniqueness of self-love in Chinese culture, it does not provide a sufficiently strong argument for why existing self-esteem or self-love scales (from Western research) are inadequate. The study primarily builds on prior qualitative research, but it does not propose a fundamentally novel theoretical framework.
While the research is important for Chinese psychology, it lacks a discussion on how the findings might be generalized or adapted for non-Chinese populations. Given that self-love is a universal human experience, failing to contextualize it within broader cross-cultural perspectives weakens its applicability.
The objectives do not explicitly mention how this scale will be applied in real world.
The study follows a standard scale development process. But needs to acknowledge in the following bias (add a Limitation section into the Discussion):
Comments 1: Sampling bias: The study relies on snowball sampling through WeChat, which may introduce selection bias, limiting generalization.
Reply 1 :Thank you for pointing that. I add the limitation section into the discussion in R417-424.
Comments 2:Ethical approval vagueness: The manuscript states IRB approval but does not specify the reference number or detailed ethical considerations.
Reply 2 : My certificate of IRB is H21044. Project Title is Theoretical Construction and Practical Path Research of Community Psychology in New Era China.The project is conducted strictly according to related law and regulations of China, and the engaged methods are invasive to participants. This Project has been supervised and approved by Institutional Review Board of Faculty of Psychology, southwest University.
Comments 3:
Reply3:During the first round of evaluation, items rated below 4.0 in relevance by 10 psychology professionals were removed. This threshold follows best practices in scale development, as expert ratings are commonly used to ensure content validity (Dewelis, 2016). Additionally, items with unclear wording were revised for clarity. In the second round, items with slightly lower scores were further reviewed and modified based on discussions with experts and evaluators. A final check was conducted to identify and eliminate any redundant items.
Comments 4:It needs more justification on the five-factor model. Why were these five factors chosen, and how do they compare with prior self-love/self-esteem models?
Reply 4:The five-factor model was derived from a rigorous qualitative study, ensuring that it captures essential dimensions of self-love in our specific cultural context. Compared to previous self-love and self-esteem models (e.g., Rosenberg, 1965; Neff, 2003), our model introduces a broader perspective by integrating self-cherishing, self-acceptance, self-restraint, self-responsibility, and self-respect. While existing models emphasize individual self-worth, our model acknowledges both personal and relational dimensions, which aligns with Confucian traditions. Furthermore, prior studies (Xue et al., 2020; Xue et al., 2021) have validated these factors through content analysis and in-depth interviews, supporting the conceptual robustness of this framework.
Comments 5:Lack of cross-validation: The study does not include a test-retest reliability analysis, which raises concerns about the longitudinal stability of the scale.
Reply 5:Thank you for pointing that. I also did the test-retest reliability, I added in R262-265: After six weeks, the test-retest reliability was assessed using the correlation coefficient between the two measurement points from the 163 collected samples. The overall correlation was 0.66, indicating moderate stability of the scale over time.
Comments 6:Insufficient external validation: The study compares its scale with some Western self-love/self-esteem scales, but it does not include real-world behavioral validation. For example, how does the scale correlate with actual self-care behaviors, mental health outcomes, or resilience?
Reply 6:Thank you for your valuable comment. In the final part of the external validity analysis, we examined the relationship between self-love and health-promoting lifestyles, which we believe provides some level of real-world behavioral validation. However, we acknowledge that this aspect could be further strengthened. Your suggestion is highly valuable for our future research, and we plan to conduct empirical studies exploring the associations between self-love, self-care behaviors, mental health outcomes, and psychological resilience. This will be explicitly addressed in the future research directions section.
Comments 7: The practical implications are weakly discussed. How can this scale be applied in mental health assessments, therapy, or educational settings?
Reply 7:Thank you for raising this point. We appreciate your suggestion regarding the insufficient external validation of the scale. In our study, we have already explored the relationship between self-love and health-promoting lifestyle behaviors, which serves as an initial form of external validation. Specifically, we examined how self-love is linked to individuals' engagement in behaviors that promote physical and mental well-being, which is an important aspect of validating the scale in real-world contexts, but we added in future research.
Comments 8:However, we acknowledge that further empirical validation with broader measures such as mental health status or psychological resilience would strengthen the external validity of the scale. We plan to explore these areas in future research to further substantiate the scale's applicability in different real-world settings.
Reply 8:Since self-love is a fundamental component of a well-rounded personality, this scale can be applied in character education practices within the Chinese cultural context. It provides a structured framework for fostering self-awareness, self-acceptance, and emotional resilience, making it valuable for educational settings, psychological counseling, and mental health interventions. In psychological assessments, it can help identify individuals struggling with low self-esteem or emotional distress. In clinical and therapeutic contexts, it may serve as a diagnostic tool to guide interventions aimed at enhancing self-compassion and self-worth. Furthermore, in educational environments, this scale can be integrated into school-based mental health programs to support students' psychological development.
We will further elaborate on these applications in the discussion section in Part 5.
Comments 9:It needs to address potential limitations in generalizing results across different Chinese subcultures (e.g., urban vs. rural, generational differences).
Reply 9:I have reanalyzed this section of the data and made the following additions: An independent samples t-test indicated that there was no significant difference in overall self-love scores between genders (t(928) = 1.661, p = 0.198). Meanwhile, a one-way ANOVA revealed a significant difference in self-love scores across different age groups (F = 3.229, p = 0.007), suggesting that age has a meaningful impact on self-love (p < 0.05).
Comments 10:Grammatical errors and awkward phrasing appear throughout. "Finally, we discussed the inheritance and development of self-love among Chinese people." → “We then analyzed how self-love is inherited and developed among Chinese adults.”
Inconsistent use of tenses (e.g., past vs. present).
Reply 10:Thank you for pointing this. I found a professional English major to check the whole manuscript.
Comments 11:Overuse of direct translations from Chinese concepts without proper contextualization for an international audience.
Reply 11:Thank you for pointing this. I translated Ren( is a fundamental concept in Confucian philosophy, often translated as "benevolence," "humaneness," or "virtue." ), Siku Quanshu(The Complete Library of the Four Treasuries: It is a large-scale collection of documents compiled during the Qianlong period of the Qing Dynasty, covered important ancient Chinese classics. It is equivalent to a systematic collection of ancient Chinese knowledge and is regarded as an important heritage of Chinese culture ).
Comments 12:The discussion section repeats many points from the results section.
Reply 12:Thank you for pointing this. I rewrote the part of discussion from the two perspective:The Inheritance and Development of Self-Love and Cross-Cultural Differences and Commonalities
Comments 13:The introduction is too lengthy, with excessive theoretical background.
Reply 13:I rewrote the introduction.
Comments 14:The results and discussion sections blend together, making it difficult to distinguish raw data from interpretation.
Reply 14:I rewrote the discussion.
Comments 15:The tables are not well explained. Some factor loadings could be visualized better (e.g., using a path diagram for the factor structure).
Reply 15:Yes,I add the figure.
Line-by-Line Comments
Line 9. "Self-love is a prominent and preeminent quality..." => Redundant phrasing. Use either "prominent" or "preeminent."
Reply: Thank you for pointing that. I revised it.
Line 18. “The Klenbach’s α..." => It should be "Cronbach’s α". Please revise it throughout the manuscript.
Reply: Thank you for pointing that. I revised it and checked the whole manuscript.
Line 19. "The confirmatory factor indicators were above 0.90." =>Specify which indicators (CFI, TLI, RMSEA).
Reply: Thank you for pointing that. I revised it and checked the whole manuscript. Such as R18-20: Thank you for pointing that. I revised it in abstract. The confirmatory factor indicators (CFI ,TFI & RESMA) were 0.91, 0.90 and 0.048.
Line 39. "They equate self-love with self-esteem..." => False generalization. Some psychologists differentiate self-love and self-esteem.
Reply:Thank you for pointing that. I revised it in R
Line 79. Please explain the concept “Sikuquanshu” and add citations.
Reply:Thank you for pointing that. Please see Reply 11.
Line 105. Table 1. is unclear in explicitly linking each hypothesis to its corresponding theoretical basis. The hypotheses are not explicitly numbered or visually aligned with their theoretical bases. The theoretical basis is provided as a block of text without clear demarcations. The relationship between hypothesis and example citations is implied rather than explicitly stated. Please use the format:
Reply: Thank you for pointing that, you gave me an excellent opinion. I have made some additions to the original style you provided. Please refer to the table 1 for details.
Table 1. Hypotheses of the Chinese Adult Self-Love Scale (SLS).
Hypothesis |
Theory Basis (Siku Quanshu) |
Supporting Quotes |
Theory Basis (Interviews with Contemporary Individuals) |
Supporting Quotes |
Hypothesis 1: Self-love includes at least the indicator of self-cherishing. |
The ancient people's care and cherishing of themselves and everything related to their existence, such as life, body, property, future, and status. |
The people who did not live up to their anticipated life expectancy or suffered from premature deaths were those who showed no caring toward themselves. (“Vol. 32,” Seven Lots from the Bookbag of the Clouds) |
The modern adults were cherishing, protecting, caring for, and respecting oneself and others |
Of course, there are physical issues to consider. This kind of care, attention, and maintenance of health is also a manifestation of self-love. If I love myself, then I am physically and mentally fine—healthy and balanced (z003). |
Line 132. "The participants could transmit the questionnaire and receive extra compensation." => Clarify: How to get compensation? Could this bias responses?
Line 170. "SPSS was used ..." => Mention version number of SPSS (e.g., SPSS 26.0).
Reply: Thank you for pointing that, I check the whole manuscript.
Several references are incomplete (e.g., missing DOI, page numbers).
APA format => needs to change to the Journal format.
Round 2
Reviewer 1 Report
Comments and Suggestions for Authors
The authors have made consistent and refined improvements to the manuscript. They have responded to each comment by providing additions, reformulations and relevant comments that add value to their work. The introduction is more carefully reformulated and more convincingly motivates this research endeavor. And the general discussions better highlight the results of the study, with pertinent and refined comments. New tables with statistical data are added that give clarity and credibility to the complicated task of building a psychological instrument.
Comment 1:
The numbering of the tables should be revised. Table no. 3 was skipped. Table no. 7 appears twice.
Comment 2:
Answer 24 is not found in the limits section of the manuscript. The 3 points should be added.
I can't comment on the quality of the English language.
Author Response
Thank you very much for your thorough review. Your insightful comments have been extremely valuable in improving my manuscript. I sincerely appreciate your time and effort.
Comment 1:
The numbering of the tables should be revised. Table no. 3 was skipped. Table no. 7 appears twice.
Reply1:Thank you very much for your careful review of the manuscript. I have already corrected the numbers in the table titles in blue.
Comment 2:
Answer 24 is not found in the limits section of the manuscript. The 3 points should be added.
Reply 2:Thank you very much for your comment. I added Rows 406-418.
This study has certain limitations that should be acknowledged. Future research should consider several directions to address these limitations and further explore the topic.
Self-report bias: Participants may be influenced by social desirability bias, providing responses that align with societal norms rather than their true self-assessment.
Impact of COVID-19: The pandemic may have affected self-love perceptions and emotions. Future studies could explore post-pandemic changes or conduct longitudinal research on its long-term impact.
Other self-concept. Moreover, current research has yet to compare self-love with related concepts such as self-esteem, self-compassion, and narcissism within a cultural context. In the future, we will use this scale as a foundation to conduct comparative studies on self-love and these related constructs, as well as to explore its applications in psychological well-being, intervention strategies, and beyond.
Reviewer 2 Report
Comments and Suggestions for Authors
The authors responded adequately all of my comments, I think the manuscript is improved now. I just have some minor comments.
- Line 76-79: consider moving this paragraph to the IRB Approval section (Line 422).
- Line 215: please cite a reference for "Questionnaire Star sample service".
- Line 249: "Gpower" should be GPower or G*Power.
Author Response
Thank you very much for your thorough review. Your insightful comments have been extremely valuable in improving my manuscript. I sincerely appreciate your time and effort.
Comment 1:Line 76-79: consider moving this paragraph to the IRB Approval section (Line 422).
Reply 1:Thank you for your comment, I moved the IRB Approval to line Rows 426-429
Comment 2: Line 215: please cite a reference for "Questionnaire Star sample service".
Reply 2:Thank you for your comment. I add the website of the“Questionnaire Star website”in Rows 494-497:
Questionnaire Star. (2020).https://www.wjx.cn/newwjx/manage/myquestionnaires.aspx?randomt=1743226334
Preliminary Test of the Chinese Adult SLS (2020-02-19):https://www.wjx.cn/hm/15vftp90u7yncirprc8w.aspx#
Expert Assessment (2020-10-22):https://www.wjx.cn/m/94580127.aspx#
Reliability and Validity(2020-11-20):https://www.wjx.cn/m/98016802.aspx#
Comment 3:Line 249: "Gpower" should be GPower or G*Power.
Reply 3: Thank you very much for your careful review of the manuscript. I revised the Gpower to G*power 3.1 in line 248
